# RNA-Based Detection of Gene Fusions in Formalin-Fixed and Paraffin-Embedded Solid Cancer Samples

**DOI:** 10.3390/cancers11091309

**Published:** 2019-09-05

**Authors:** Martina Kirchner, Olaf Neumann, Anna-Lena Volckmar, Fabian Stögbauer, Michael Allgäuer, Daniel Kazdal, Jan Budczies, Eugen Rempel, Regine Brandt, Suranand Babu Talla, Moritz von Winterfeld, Jonas Leichsenring, Tilmann Bochtler, Alwin Krämer, Christoph Springfeld, Peter Schirmacher, Roland Penzel, Volker Endris, Albrecht Stenzinger

**Affiliations:** 1Institute of Pathology, University Hospital Heidelberg, 69120 Heidelberg, Germany; 2German Cancer Consortium (DKTK), 69120 Heidelberg partner sites, Germany; 3Clinical Cooperation Unit Molecular Hematology/Oncology, German Cancer Research Center (DKFZ), 69120 Heidelberg, Germany; 4Department of Internal Medicine V, University Hospital Heidelberg, 69120 Heidelberg, Germany; 5Department of Internal Medicine VI, University Hospital Heidelberg, 69120 Heidelberg, Germany; 6National Center for Tumor Diseases (NCT), 69120 Heidelberg, Germany

**Keywords:** gene fusion, targeted therapy, carcinoma, solid tumor, NGS

## Abstract

Oncogenic gene fusions are important drivers in many cancer types, including carcinomas, with diagnostic and therapeutic implications. Hence, sensitive and rapid methods for parallel profiling in formalin-fixed and paraffin-embedded (FFPE) specimens are needed. In this study we analyzed gene fusions in a cohort of 517 cases where standard treatment options were exhausted. To this end the Archer^®^ DX Solid tumor panel (AMP; 285 cases) and the Oncomine Comprehensive Assay v3 (OCA; 232 cases) were employed. Findings were validated by Sanger sequencing, fluorescence in situ hybridization (FISH) or immunohistochemistry. Both assays demonstrated minimal dropout rates (AMP: 2.4%; *n* = 7/292, OCA: 2.1%; *n* = 5/237) with turnaround times of 6–9 working days (median, OCA and AMP, respectively). Hands-on-time for library preparation was 6 h (AMP) and 2 h (OCA). We detected *n* = 40 fusion-positive cases (7.7%) with TMPRSS2::ERG in prostate cancer being most prevalent (*n* = 9/40; 22.5%), followed by other gene fusions identified in cancers of unknown primary (*n* = 6/40; 15.0%), adenoid cystic carcinoma (*n* = 7/40; 17.5%), and pancreatic cancer (*n* = 7/40; 17.5%). Our results demonstrate that targeted RNA-sequencing of FFPE samples is feasible, and a well-suited approach for the detection of gene fusions in a routine clinical setting.

## 1. Introduction

Cancer is primarily a genetic disease. Besides point mutations, small insertion and deletion (InDels) and copy number alterations, chromosomal translocations resulting in hybrid oncogenic gene fusions play a crucial role in tumor development. Fusion genes were first discovered in hematologic malignancies, such as *BCR-ABL1* in chronic myelogenous leukemia [1] or *MYC-IGH/IGK/IGL* in Burkitt’s lymphoma [2]. There, easy access and in vitro cultivation of tumor material from blood led to the discovery of inter-chromosomal translocations in near-to-diploid karyograms [3]. Since then, technical advances facilitated detection of fusion genes also in solid tumor entities. It turned out that many sarcomas especially harbor specific fusion genes that now can be utilized to differentiate between sarcoma entities, for example, the *EWSR1-FLI* fusion gene for the differential diagnosis of small round blue cell tumors [4]. Other fusions like EWSR1::NFATc2 have been used to define new (sub-) entities with different genetic, epigenetic, transcriptomic and epidemiological profiles [5], despite the same histomorphological appearance [6].

While gene fusions serve as characteristic marker lesions in many hematological and mesenchymal tumors, they also have only more recently been unveiled as oncogenic drivers in and across various carcinoma subtypes and can be utilized for diagnostic purposes as well as therapeutic interventions. For example, clinically exploitable chromosomal fusion genes involving *ALK*, *ROS1*, *RET*, or *NTRK* [7] are well described for lung adenocarcinoma. In other solid tumor entities, such as gastrointestinal stromal tumor (GIST) [8], breast cancer [9,10] or tumors of the central nervous system [11], targetable gene fusions at variable frequencies have been identified [11,12]. From a diagnostic perspective, major obstacles delaying a comprehensive delineation of the landscape of gene fusions [13] and their full clinical exploitation are inherent limitations of current detection methods and the inapplicability of comprehensive sequencing approaches (i.e., whole genome (WGS) and full transcriptome sequencing) in a large-scale diagnostic outreach setting. Economic considerations, as well as the widespread diagnostic use of formalin-fixed and paraffin-embedded (FFPE) tumor specimens often containing heavily degraded nucleic acids and substantial variability in tumor cellularity, are key determinants in this regard. Bridging this gap, several targeted sequencing solutions for the detection of gene fusions have been developed recently [14,15,16], but reports of large scale applications in a clinical setting are still limited [17,18]. However, this real-world evidence across many centers and institutions will be crucial in guiding future diagnostic avenues and standards.

In this study, we analyzed the performance and characteristics of two RNA-based targeted next generation sequencing assays in a routine clinical setting and describe the gene fusion events observed in FFPE cancer samples.

## 2. Results

We analyzed targeted RNA sequencing data of 517 FFPE tumor samples from patients with different solid cancer types other than non-small cell lung cancer, who progressed beyond standard of care. An overview of the study outline, the performed analyses and a summary of the results are visualized in Figure 1. Due to different clinical settings, there were two sub-cohorts. In case of requests for gene fusion analysis only, analysis was performed with the Archer^®^ DX Fusion Plex^®^ Solid tumor panel based on anchored multiplex polymerase chain reaction (PCR) [19] (AMP; AMP-cohort, *n* = 285 patients). The Oncomine™ Comprehensive Assay v3 [20,21,22] (OCA, a combination of DNA- and RNA-targeted sequencing, was used for molecular profiling beyond fusions; OCA-cohort, *n* = 232 patients) (Table 1). Gene fusions detected by these assays were further analyzed by orthogonal Sanger sequencing, immunohistochemistry or fluorescence in situ hybridization (FISH) (Table 2 and Appendix A).

### 2.1. Cohort Characteristics, Tumor Cell Content and Sample Dropouts

Age was comparable with a median of 62.1 years for the AMP cohort and 60.4 years for the OCA cohort. The AMP cohort harbored fewer females with 38.2% in contrast to 54.7% in the OCA cohort. Eighteen different tumor entities were analyzed with the AMP assay and 14 with the OCA panel (for details see Table 1 and Figure 2A,B). Median tumor purity of macrodissected tissue was high (AMP: 80%, OCA: 70%; range: 10 to 95%). Dropout rates were minimal with 2.4% (7/292) and 2.1% (*n* = 5/237) for the AMP cohort and the OCA cohort, respectively. Causes for dropouts were low RNA quality (AMP: 1.7%, *n* = 5; OCA: 1.7%, *n* = 4) and tumor cellularity below 10% (AMP: 0.7%, *n* = 2; OCA: 0.4%, *n* = 1).

### 2.2. Turnaround Time and RNA Yields

Turnaround times (TAT), which we defined as the time that elapses from receipt of material in our laboratory to reporting, were low for both assays; 9 working days (median) for the AMP-cohort and 6 working days (median) for the OCA-cohort (Table 1). As described in Figure 1 the hands-on-time for the library preparation per AMP sample was about 6 h with a total time of 12 h for the assay workflow compared to about 2 h hands-on-time and a total time of 6 h per OCA sample. Since for most cases of the AMP cohort surgical resection material was available, the median RNA yield generated by the Maxwell^®^ 16 Research extraction system was higher for AMP (41.5 ng/µL) than for OCA (29.3 ng/µL) samples. Of note, some cases comprised only minimal amounts of RNA (as low as 2.4 ng/µL). The amount of template RNA used for library preparation was either 20 ng for OCA or at least 100 ng for the AMP assay.

### 2.3. Detection and Validation of Gene Fusions

In total, we detected gene fusions in 40 cases (7.7%) out of the entire cohort (Figure 1 & Table 2). In six of these cases fusions (*n* = 3 with AMP: AXL::CAPN15, GPBP1L1::MAST2, MTMR::MAML2 and *n* = 3 with OCA: FNDC3B::PIK3CA, KIF5B::RET, TBL1XR1::PIK3CA) were only detectable with low abundance of transcripts (ranging from 5 to 744 reads), and could not be confirmed by Sanger sequencing. These cases were considered discordant.

Next, we analyzed the 6 discordant cases further by performing a cross-comparison between both RNA sequencing assays. Due to limited sample material (case: #OCA-6) or assay limitations (cases: #AMP-1, #AMP-6 and #AMP-8: the OCA panel did not contain primers for the fusions detected by AMP), this analysis was only possible for two samples (cases: #OCA-5 and #OCA-9). The respective gene fusions detected by the OCA panel (FNDC3B::PIK3CA and TBL1XR1::PIK3CA) were not identified by the AMP assay.

In a third step, we cross-tested a total of 12 fusion-positive samples originally identified by either one of the assays (*n* = 6 cases each) with the respective other approach. The gene fusions PTPRK::RSPO3, SND1::BRAF, TMPRSS2::ERG and WHSC1L1::FGFR1 originally detected by the AMP assay (cases: #AMP-16, #AMP-17, #AMP-19, #AMP-21, #AMP-24 und #AMP-27) were all identified by the OCA panel. Of the 6 gene fusions (BRD4::NUTM1, SND1::BRAF, TMPRSS2::ERG, TRIM24::BRAF and WHSC1L1::FGFR1) detected with OCA (cases: #OCA-1, #OCA-8, #OCA-10, #OCA-11, #OCA-12, and #OCA-13), only 4 (cases: #OCA-1, #OCA-8, #OCA-11 and #OCA-12) were also identified by the AMP approach. These fusions TBL1XR1::PIK3CA and WHSC1L1::FGFR1 (#OCA-10 and OCA-13) that were not detected by AMP showed only a low amount of transcript reads (973 and >345) in the OCA assay.

The AMP detection system entails a quality check of FFPE samples prior library preparation. To this end, a quantitative polymerase chain reaction (qPCR)-based method (PreSeq RNA Quality Control, Figure 1) was used to determine the quality of the nucleic acids present in the sample material. The PreSeq score that was determined by this step is a relative measure of the target concentration in the PCR reaction. This information can be used to optimize input based on tumor cellularity or to withhold poor quality samples from library preparation. In our experience, PreSeq scores > 28 are a clear indication of poor nucleic acid quality. To test drop-outs from one assay on the other, we analyzed three samples with a PreSeq score > 28. While these cases were not suited for the AMP assay, the OCA assay was run successfully. In these three cases, no gene fusions were detectable.

Lastly, we cross-tested a total of 10 negative fusion samples identified by either AMP or OCA (*n* = 5 each). All 10 cases were identified as fusion-negative by both assays.

### 2.4. Gene Fusions in Different Cancer Types

An overview of the different cancer types with a gene fusion confirmed by an orthogonal method is shown in Figure 2C,D for both cohorts. In the AMP cohort *n* = 24 confirmed fusion samples were detected (Figure 3A) in seven different entities (prostate cancer 37.5%, *n* = 9/24; adenoid cystic carcinoma 29.17%, *n* = 7/24, pancreatic cancer 16.7%, *n* = 4/24; cholangiocarcinoma 4.17%, *n* = 1/24; lung cancer 4.17%, *n* = 1/24; neuroendocrine tumor of the pancreas 4.17%, *n* = 1/24; mammary analogue secretory carcinoma (MASC; according to the new World Health Organization (WHO) classification: secretory carcinoma of the salivary gland) 4.17%, *n* = 1/24). In the OCA cohort *n* = 10 confirmed fusion cases were identified (Figure 3B) in 4 different entities (CUP 60%, *n* = 6/10; pancreatic cancer 20%, *n* = 2/10; lung cancer 10%, *n* = 1/10 and thyroid carcinoma 10%, *n* = 1/10).

Recurrent gene fusions were primarily identified in prostate cancer (TMPRSS2::ERG) and adenoid cystic carcinoma (MYB::NFIB) in 1.74% (9/517) and 1.35% (7/517), respectively in the overall cohort. Interestingly, the gene fusion WHSC1L1::FGFR1 was detected in one patient each from the AMP and the OCA cohort (adenocarcinoma of pancreatobiliary type and anaplastic thyroid carcinoma, respectively). Of both *TMPRSS2* and *WHSCL11*, the non-coding 1st exons were fused to their respective partners thus indicating that the gene fusion would lead to the overexpression of the fusion partner under a different promoter than normally. The gene fusion SND1::BRAF that was previously described in pancreatic acinar carcinoma [23] was detected in two cases of pancreatic ductal adenocarcinoma (sequenced with AMP and OCA, respectively).

### 2.5. Clinical and Diagnostic Implications of Gene Fusions

We identified a total of 34 cases with gene fusions that have either therapeutic or diagnostic implications (6.6% of the entire cohort, 85.0% of the fusion-positive cases). As detailed in Appendix A, we observed 12 cases where gene fusions are potentially druggable in clinical trials or experimental clinical settings and another 23 cases where the detected gene fusion can support routine diagnostics.

### 2.6. Gene Fusion Isoforms

A detailed analysis of the fusions detected with AMP revealed that all different fusion isoforms (fractions) were clearly identified and separated by the use of molecular barcodes for each transcript. Notably, in all TMPRSS2::ERG fusions present in prostate cancer, we identified more than one fusion splicing isoform (Figure 4A,B). We also identified such isoforms in case #OCA-11 (prostate cancer that was previously diagnosed as a CUP) using the OCA panel. In comparison, Sanger sequencing data from samples containing more than one mRNA fraction starting from the same exon, for example, by splicing, did not provide analyzable results (Figure 4C,D), illustrating the advantage of next generation sequencing based AMP over Sanger sequencing.

### 2.7. Identification of Previously Unknown Gene Fusions

Due to the methodology [19], the AMP assay is able to identify gene fusions harboring unknown fusion partners. Interestingly, we were also able to identify previously unknown gene fusions with the OCAv3 RNA panel as the RNA panel contains approximately 950 primer combinations for known gene fusions, which can recombine if primers for the potentially fused genes are present in the panel. During our analysis we encountered 4 fusions (ESR1::QKI, RNF130::SEPT14, TRIM24::BRAF and BRD4::NUTM1), for which the calling algorithm identified “novel” fusion transcripts gained from primer combinations that were not initially designed as pair. For example, the EGFR::SEPT14 and the RNF130::BRAF fusions are known from literature, but within the OCA cohort we identified an RNF130::SEPT14 fusion. Yet another example is a BRD4::NUTM1 fusion, harboring a fusion point in Exon 2 of *NUTM1* which is not targeted with the original primer combination but rather with a primer downstream in Exon 2 of *NUTM1*, binding at a position which has been described with WHSC1L1::NUTM1. These gene fusions were all confirmed by Sanger sequencing.

### 2.8. Individual Cases

A classic MASC case is shown in Figure 5A (case: #AMP-3). Here, we identified an ETV6::NTRK3 fusion transcript using the AMP assay. The RNA sequencing data indicated that this gene fusion was in-frame and the functional NTRK3 tyrosine kinase domain was intact. We validated this result by Sanger sequencing and additionally employed a pan-neurotrophic-tropomyosin receptor tyrosine kinase (NTRK) antibody to verify overexpression of NTRK on the protein level. Another case showed a pancreatoblastoma with an FGFR2::INA [fusion (Figure 5B, case: #AMP-4). This gene fusion has been described in mixed neuronal-glial tumors [24], but has not yet been described in pancreatoblastoma to our best knowledge. The functional tyrosine kinase domain was intact. Fusion with the n-terminal internexin neuronal intermediate filament protein alpha (INA) domain potentially leads to dimerization without ligand binding and activation of mitogen-activated protein kinase (MAPK) signaling, which might be clinically exploitable [23]. As shown in Figure 5C, case #OCA-1 had a NUT midline carcinoma, a diagnosis that was supported by immunhistochemical detection of NUT. The OCAv3 RNA panel successfully identified *NUT* fused to *BRD4*. 

## 3. Discussion

In this study we show that gene fusions can be detected by targeted RNA-based next-generation sequencing in FFPE carcinoma samples other than of lung origin [17,18,25] in a routine diagnostics set-up. Based on our quality-controlled and well established workflows with FFPE-derived RNA [25], we were able to analyze and identify gene fusions in samples containing only low (<3.0 ng/µL) and strongly degraded RNA with two different targeted RNA-sequencing assays. 

By design [19], the AMP panel facilitates the detection of yet unknown gene fusions. This is due to amplicons that are generated by a 3’-RACE PCR, which allows amplification of sequences from an anchored primer into the “unknown”. Hence, known oncogenic kinases like tyrosine-protein kinase Met (MET), serin/threonin-kinase B-Raf (BRAF), or neurotrophic-tropomyosin receptor tyrosine kinase (NTRK) [26,27], which may fuse with various partners, are the most suited candidates for the AMP assay. Other examples are cases with a genomic 3’-5’ imbalance in targeted gene fusion panels that do not cover the gene fusion by their design or cases where a translocation is detected by break-apart FISH but requires identification of the actual gene fusion. The amount of RNA needed (200 ng per sample, min. 100 ng) is high (approximately 10 times higher compared to the OCA panel) due to a double cDNA synthesis step.

The amplicon-based OCA panel contains primers covering a wide range of the most prominent exon combinations and is primarily focused on the identification of known genes fusion. Nonetheless, we here show that analysis of new combinations of the existing primer pairs can lead to the identification of fusions beyond the ones for which the assay was originally designed. The amount of RNA necessary for the OCA assay is 20 ng, but in our experience as little as 10 ng can be sufficiently analyzed. This can be of particular help for the analysis of small samples like biopsies [18].

The entire work-flow from receipt of the sample to reporting led to a total median TAT for OCA samples of 6 and for AMP samples of 9 working days. The TAT reported here may vary as it is not only influenced by assay-specific requirements but also by the specific institutional set-up and workflow [18]. The main influencing factors in our study can be summarized as follows: (i) if a sample does not fulfill the quality control standards we try to obtain alternative material form external partners which is then used for analysis; and (ii) due to the different molecular barcoding systems of the OCA- and the AMP-panel, panels must be run on separate sequencing chips. To optimize cost-value ratio we pool samples for a maximum of four days until an ideally fully loaded chip is run. In summary, we were able to perform both analyses with a TAT of less than 10 working days, which is currently recommended by national and international guidelines for molecular diagnostics of non-small cell lung cancer [28,29].

Six of the 40 gene fusions detected could not be confirmed by Sanger sequencing. All shared a low abundance of transcripts (ranging from 5 to 744 reads) probably leading to absent PCR amplicons and subsequent Sanger sequences. While for this study we defined Sanger sequencing as a gold standard, we acknowledge that in these discordant cases, Sanger sequencing may lack sensitivity, particularly in low tumor content or low tumor cellularity. These data indicate that additional studies on this topic are needed, which will collectively lay the foundation for new diagnostic standards in the detection of gene fusions from clinical samples.

In our study, we detected 12 (30.0%) gene fusions that are potentially druggable, such as FGFR2::INA, VTCN1::NRG1, and EML4::ALK. Further 23 (57.5%) gene fusions, such as TMPRSS2::ERG or MYB::NFIB, can support differential diagnostics (for a summary see Appendix A). For example, in one case that initially presented as a CUP, a TMPRSS2::ERG fusion could be identified. Careful re-evaluation of the histopathology in context of the molecular findings revealed that the tumor was indeed a prostate cancer and the patient was treated accordingly. This finding is in line with recent observations by another group, who reported an index case of prostate cancer with variable histology that was only fully diagnosed by the identification of TMPRSS2::ERG fusion in both, the squamous (divergent histology) and in the primary prostatic adenocarcinoma specimen [30]. Interestingly, TMPRSS2::ERG fusion cases in our study showed different fractions of fusion isoforms not detectable with conventional Sanger sequencing, where mostly the non-coding 1st exon of *TMPRSS2* is fused with the *ERG* coding sequence. The consequence of this is the transcription starts from an alternative start codon within the *ERG* coding sequence leading to a truncated ERG protein which has lost its upstream regulatory domains [31,32].

In summary, we here provide evidence that RNA sequencing for the detection of gene fusions in FFPE carcinoma samples is a reliable and fast tool that can be integrated in routine diagnostics. Screening of gene fusions beyond known fusion-positive cancer types can inform tumor biology, diagnosis and clinical therapy.

## 4. Material and Methods

### 4.1. RNA Extraction, Library Preparation and Semiconductor Sequencing

RNA extraction, library preparation, and semiconductor sequencing were performed as described previously [25,33]. A summary of the entire workflow is depicted in Figure 1. In short, tumor tissue was macrodissected to achieve a histological tumor cell content of at least 15%. The extraction was carried out with the automated Maxwell^®^ 16 Research extraction system (Promega, Madison, WI, USA) and the Maxwell^®^ 16 FFPE Plus LEV RNA Purification Kit following the manufacturer’s instructions. The samples were then digested with the TURBO DNA-free ™ Kit (Thermo Fisher Scientific, Waltham, MA, USA) to obtain DNA-free RNA. The concentration of RNA was measured fluorimetrically (QuBit 2.0 RNA high sensitivity kit (Thermo Fisher Scientific). 

### 4.2. Library Preparation for the Oncomine™ Comprehensive RNA Panel v3

For library preparation, the multiplex PCR-based Ion Torrent AmpliSeq^TM^ technology (Life Technologies, Thermo Fisher Scientific) with the Oncomine™ Comprehensive Assay v3 (IonTorrent, Thermo Fisher Scientific) was used.

For detection of gene-fusions, RNA was reversely transcribed with the SuperScript™ VILO™ cDNA Synthesis Kit according to the manufacturer’s handbook (Invitrogen, Thermo Fisher Scientific). Here, the amplicon libraries were prepared from 20 ng RNA which were mixed with two primer pools generating 1732 amplicons (Appendix A) and the AmpliSeq HiFi Master Mix and transferred to a PCR cycler (BioRad, Munich, Germany). After the end of the PCR reaction, RNA primer end sequences were partially digested using FuPa reagent, followed by the ligation of barcoded sequencing adapters (Ion Xpress Barcode Adapters; Life Technologies, Thermo Fisher Scientific). The final libraries were purified using AMPure XP magnetic beads (Beckman Coulter, Krefeld, Germany) and quantified using qPCR (Ion Library Quantitation Kit, Thermo Fisher Scientific) on a StepOne qPCR machine (Thermo Fisher Scientific).

The individual libraries were diluted to a final concentration of 50 pM and samples were pooled and processed to library amplification on Ion Spheres using Ion 510 & 520 & 530 or 540 Chef Kit and library enrichment on the Ion Chef (Thermo Fisher Scientific). The libraries were then processed for sequencing using the Ion S5 Sequencing chemistry. The barcoded libraries were loaded onto 530 or 540 chips and the samples were pooled to achieve at least 500,000 reads (250,000 reads per primer pool). For samples not reaching the minimum amount of reads per library, preparation was repeated and the sample re-sequenced. The sequence quality for each sample was assessed by the following passing criteria: at least 500,000 total reads and 5 Housekeeping Genes with at least 40,000 reads each. If all criteria were matched but no specific fusion was detected, the sample was considered negative for fusions covered by the assay specific primers. To call a fusion, at least 1000 reads of that specific fusion were required in addition to the aforementioned criteria; if the read count was lower, a second method (RT-PCR or AMP) was used for confirmation.

### 4.3. Library Preparation for AMP Based NGS (Archer Dx) Translocation Detection

For enriched cDNA library preparation, the multiplex PCR-based Ion Torrent AmpliSeq^TM^ technology (Life Technologies, Thermo Fisher Scientific) with the Archer^®^ FusionPlex^®^ Solid Tumor kit v3 (Archer^®^ DX, Boulder, CO, USA) was used according to the manufacturer’s instructions. In short, 100–200 ng RNA were reversely transcribed for first strand synthesis and subjected to real-time PCR (Archer^®^ PreSeq RNA QC assay; Archer^®^ DX). Samples with an insufficient quality control (QC) score > 28 were discarded and RNA was isolated from an alternative FFPE block (where available). Next, a second strand synthesis was performed followed by product end repair, phosphorylation and dA-tailing (adenylation) with ligation of the Ion torrent specific molecular barcode (MBC) v2 adapters resulting in half functional molecular-barcoded double-stranded cDNA. First, PCR was performed with anchored gene specific primers covering 53 target genes (Appendix A) and a universal primer located at the end of the MBC v2 adapter. The second PCR with nested gene specific primers carrying the index for sample multiplexing and the universal primers produces the fully functional library which was quantified using qPCR (Ion Library Quantitation Kit, Thermo Fisher Scientific) on a StepOne qPCR machine (Thermo Fisher Scientific). Libraries < 50,000 pM were discarded and library preparation was repeated with higher amounts of RNA. The individual libraries were diluted to a final concentration of 50 pM and samples were pooled and processed to library amplification on Ion Spheres using Ion 510 & 520 & 530 or 540 Chef Kit and library enrichment on the Ion Chef (Thermo Fisher Scientific). The libraries were then processed for sequencing using the Ion S5 Sequencing chemistry (Thermo Fisher Scientific). The barcoded libraries were loaded onto 530 (max. 5 libraries) or 540 (max. 14 libraries) chips, to achieve a minimum amount of 3 million reads per sample.

The data was analysed with the Archer^®^ analysis software (Version 5.1; Archer^®^ DX) for the presence of gene fusions. The sequence quality for each sample was assessed by the following criteria: >10% or at least 150,000 unique fragments, >50 average unique RNA start sites per gene specific primer 2 (GSP2) control and on-target deduplication ratio < 40. If all criteria were matched, but no specific fusion was detected, the sample was considered negative for fusions, which could be detected with the assay specific primers.

### 4.4. Reverse Transcription-Polymerase Chain Reaction (RT-PCR) and Sanger Sequencing Validation

The amount of 20 ng RNA served as template for target specific cDNA synthesis and subsequent PCR using the SuperScript™ VILO™ cDNA Synthesis Kit according to the manufacturer’s handbook (Invitrogen, Thermo Fisher Scientific). Primers were designed to align to either side of the specific fusion breakpoints (Appendix A). PCR product size was confirmed by agarose gel electrophoresis. PCR products were excised from the gel and cleaned up in case of multiple bands or directly Sanger sequenced in both directions on an ABI 3500 Sequencer (Applied Biosystems, Thermo Fisher Scientific).

### 4.5. Immunohistochemistry

For detection of fusion protein expression, 3 μm thick paraffin sections were prepared. Deparaffinization and tissue staining were performed using a Benchmark Ultra IHC Staining module according to standard protocols. Primary antibodies used for analysis were: anti-NUT (clone C52B1, dilution 1:25, Cell Signaling, Frankfurt, Germany), anti-TRK A-C (EPR17341, dilution 1:10, Abcam, Berlin, Germany). Staining was visualized using the Vectastain elite ABC detection system (Vector, Burlingame, CA, USA) and using 3,3′-Diaminobenzidine (DAB, Dako; Agilent, Santa Clara, CA, USA) as chromogen. Hematoxylin was used for counterstaining of cell nuclei.

### 4.6. FISH Analyses

FISH analyses were performed on whole block slides of formalin-fixed and paraffin-embedded samples. For detection of *FGFR2* rearrangements, the ZytoLight SPEC FGFR2 Dual Color Break Apart Probe (Zytovison, Bremerhaven, Germany, Prod. No.: Z-2169-200) was used according to the manufacturer’s instructions. Cases were evaluated by experienced pathologists.

### 4.7. Research Ethics

The study was conducted in accordance with the Declaration of Helsinki and local ethics regulations. A protocol was approved (S-638-2016) by the Ethics Committee of the University of Heidelberg.

## 5. Conclusions

Oncogenic gene fusions occur across many cancer types and may aid diagnosis or provide therapeutic opportunities. Targeted RNA sequencing interrogates multiple genes on transcript level simultaneously and identifies the fusion partners and exons. As shown here, successful implementation and use requires precise knowledge of the pros and cons of each assay as pre-analytic and analytic requirements and abilities vary. Communication of this information between clinical colleagues and pathologists is critically important to identify the best approach for fusion detection in a given clinical setting as well as to understand and evaluate the molecular pathology report.

## Figures and Tables

**Figure 1 cancers-11-01309-f001:**
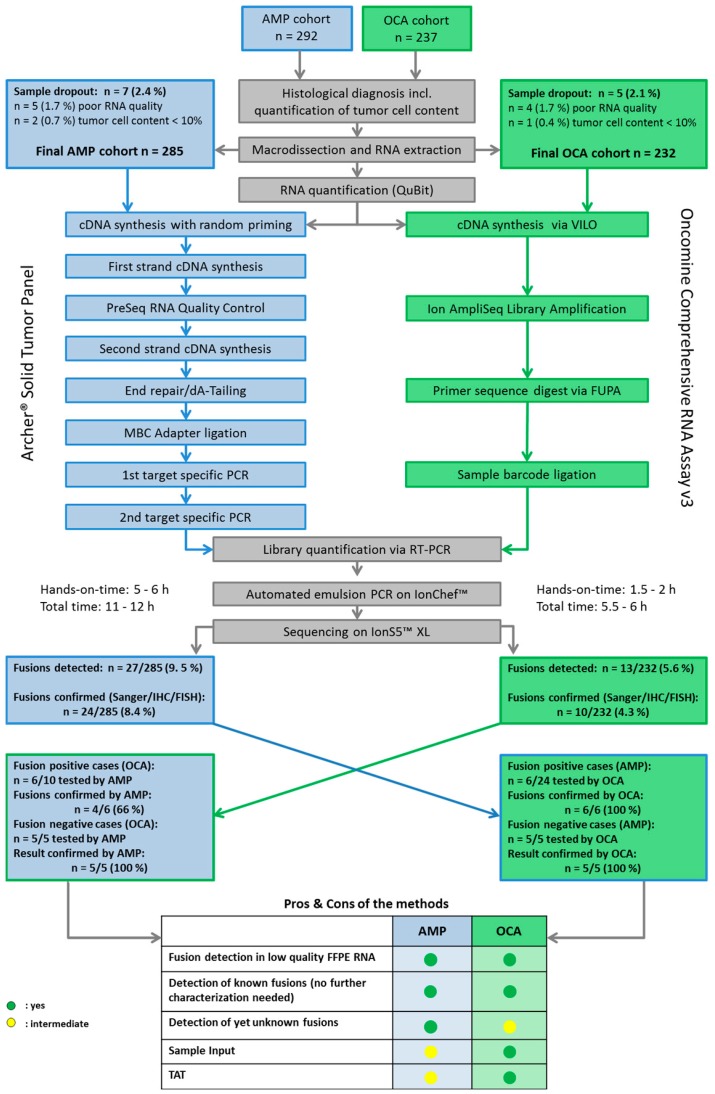
Study outline: Sample preparation and data analysis. Blue: Sample preparation steps used for the cohort sequenced with the Archer^®^ DX Fusion Plex^®^ Solid tumor panel (AMP). Green: Sample preparation steps used for the cohort sequenced with the Oncomine™ Comprehensive Assay v3 (OCA). Grey: Sample preparation steps used for both cohorts.

**Figure 2 cancers-11-01309-f002:**
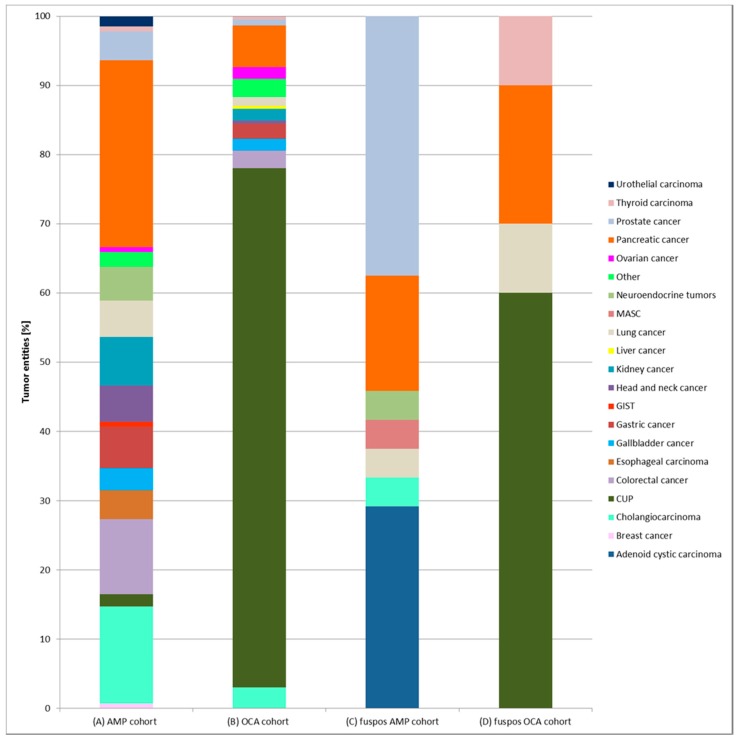
Frequencies of cancer types in the overall cohort as well as in the fusion positive cohort. (**A**) All tumor entities of the cohort sequenced with AMP. (**B**) All tumor entities of the cohort sequenced with OCA. (**C**) Tumor entities of the fusion positive cases sequenced with AMP. (**D**) Tumor entities of the fusion positive cases sequenced with OCA.

**Figure 3 cancers-11-01309-f003:**
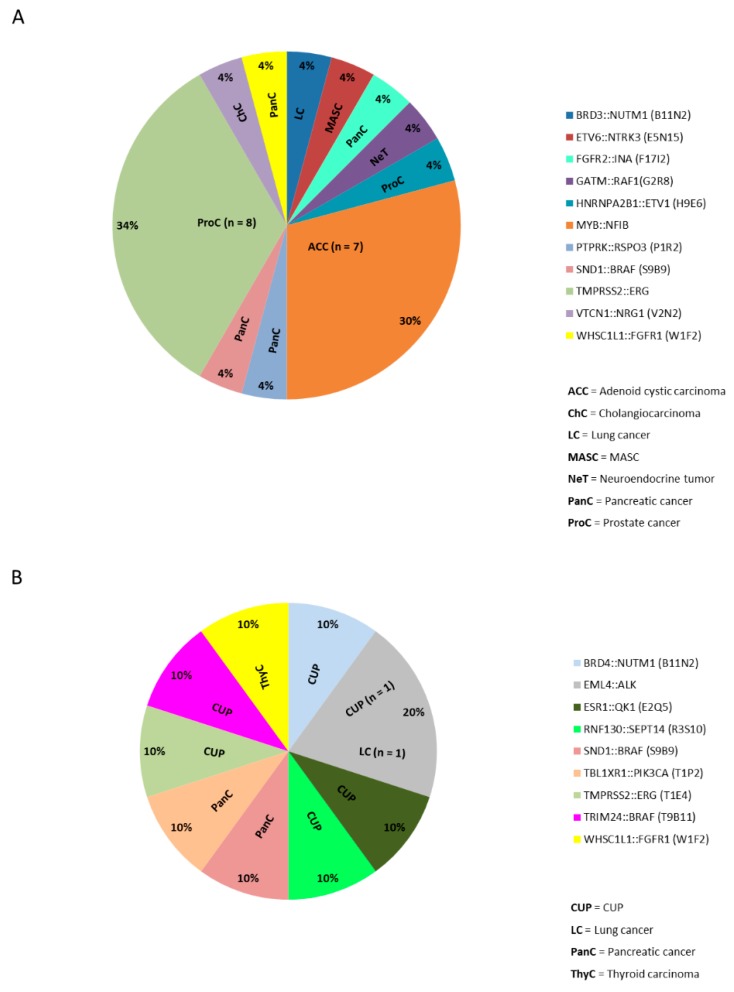
Frequency of gene fusions identified with AMP or OCA in the analyzed cancer types. Figure 3 shows the frequency of gene fusion positive cases identified with (**A**) the AMP or (**B**) the OCA assay in the different tumor entities.

**Figure 4 cancers-11-01309-f004:**
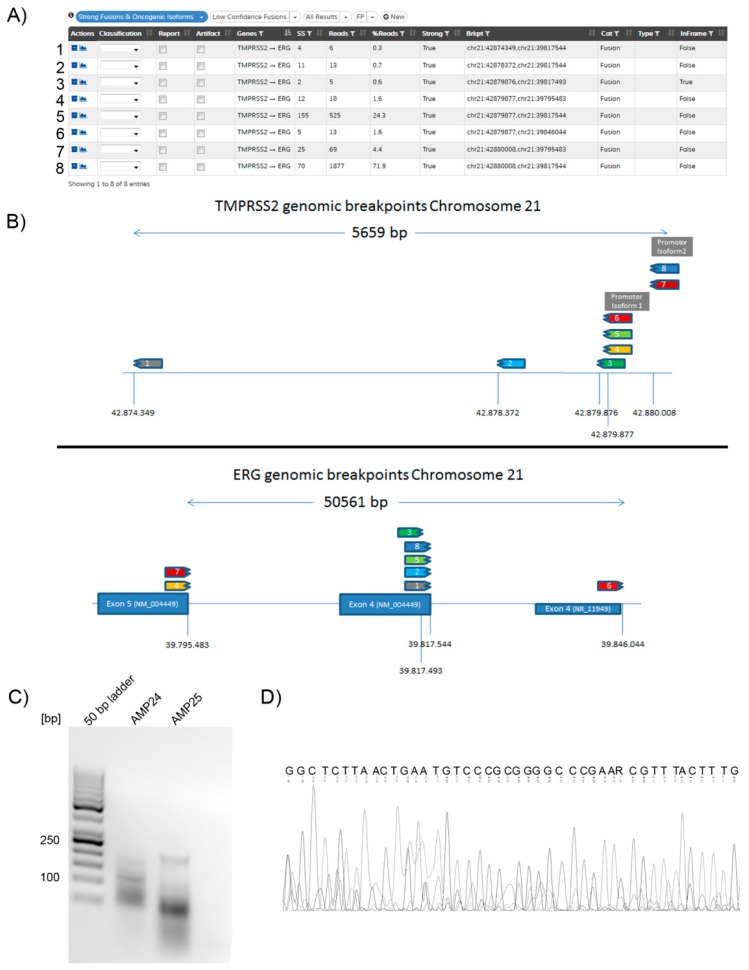
Representative results for an identified TMPRSS2::ERG fusion (case # AMP-25). (**A**) Typical Archer results screen for identified fusions. The patient #AMP-25 harbored 8 different fractions of TMPRSS2::ERG fusion transcripts. Due to molecular barcoding, all different fusion transcripts can be identified separately. (**B**) Schematic overview of the genomic breakpoints in the *TMPRSS2* and *ERG* genes. The numbering of the different fusion splicing isoforms corresponds between Figure 4A and Figure 4B. (**C**) Agarose gel: Size separation of the reverse transcription PCR (RT-PCR) products from 2 samples positive for the TMPRSS2::ERG fusion tested by AMP. The amplicon size for this primer combination is 185 bp. The prominent band of sample #AMP-25 equal to the size of 200 bp was cut out, the DNA was isolated and the product sequenced. From left to right: 50 bp ladder, RT-PCR product case #AMP-24, RT-PCR product case #AMP-25, no template control. (**D**) Sanger sequencing: Part of the sequence from the cut out band; due to the multiple splicing forms, alignment with the human genome and transcriptome gave no clear result of one specific TMPRSS2::ERG fusion sequence.

**Figure 5 cancers-11-01309-f005:**
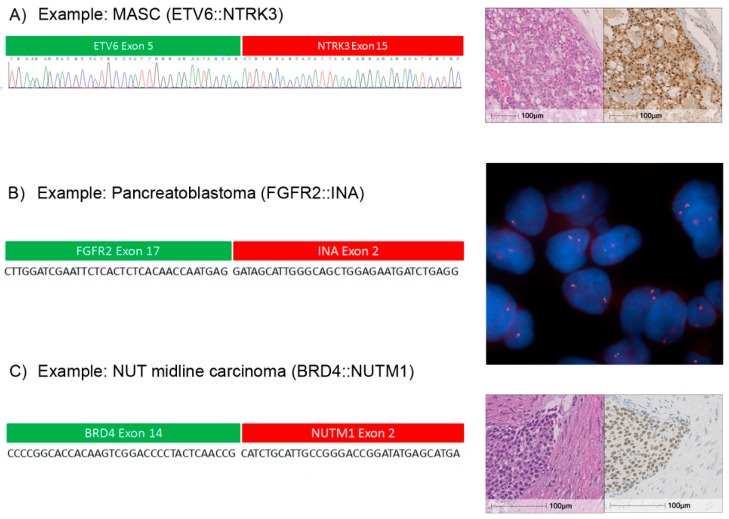
Three cases with gene fusions detected by targeted RNA-sequencing. (**A**) Case: #AMP-3: Mammary analogue secretory carcinoma carrying an ETV6::NTRK3 fusion detected by AMP. The green and red bars above the reference sequence show which part of the transcript belongs to which gene; below the Sanger sequencing result of the *ETV6* and *NTRK3* fusion transcript is shown. Microphotographs of hematoxylin and eosin stain and immunohistochemical stain for anti- tropomyosin receptor tyrosine kinase (TRK). (**B**) Case: #AMP-4, from left to right: Sequence of a typical *FGFR2* and *INA* fusion read detected by AMP; the green and red bars above the reference sequence show which part of the transcript belongs to which gene. Right: break apart detection of the fusion by FISH. (**C**) Case: #OCA-1, from left to right: Sequence of a typical *BRD4* and *NUTM1* fusion read detected by OCA in a NUT midline carcinoma; the green and red bar above the reference sequence show which part of the transcript belongs to which gene. Right: Microphotographs of hematoxylin and eosin stain and immunohistochemical stain for anti NUTM1.

**Table 1 cancers-11-01309-t001:** Characteristics of cohorts and assay features.

Characteristics and Features	All	AMP	OCA
***n***	517	285	232
**age [y]**	61.5 (4–84)	62.1 (6–84)	60.4 (4–82)
**% females**	45.7	38.2	54.7
**n different entities**	21	18	14
**tumor cell content [%]**	70 (10–95)	80 (10–95)	70 (10–95)
**Turnaround time [days]**		9 (3-35)	6 (3-44)
**RNA [ng/µL]**	35.5 (2.4–880)	41.5 (2.4–880)	29.3 (2.6–499)
**% fusion positive**	6.58	8.42	4.31
**Unique fragments > 150,000**		465,740 (21,498–3,974,814)	
**Unique fragments > 10%**		8.6 (0.7–48.4)	
**Average unique RNA start sites**		171.88 (18.6–405.3)	
**per GSPS control > 50**			
**On target deduplication ratio < 40**		12.2 (2.0–144.4)	
**Usable reads**			579,492 (37,591–13,897,936)

Given is always median with minimum and maximum values.

**Table 2 cancers-11-01309-t002:** Summary of the detected gene fusions in the overall cohort. MASC; mammary analogue secretory carcinoma. CUP; cancer of unknown primary. NUT; nuclear protein in testis.

Sample ID	Fusion	Entity	Reads	Panel	Validation Successful with Any Method *
AMP-1	AXL::CAPN15 (A19C2)	Gastric adenocarcinoma	6	AMP	no
AMP-2	BRD3::NUTM1 (B11N2)	NUT-midline carcinoma of the lung	2968	AMP	
AMP-3	ETV6::NTRK3 (E5N15)	MASC	15,730	AMP	
AMP-4	FGFR2::INA (F17I2)	Pancreatic blastoma	>100	AMP	
AMP-5	GATM::RAF1 (G2R8)	Neuroendocrine tumor of the pancreas	4186	AMP	
AMP-6	GPBP1L1::MAST2 (G6M4)	Adenocarcinoma, pancreatobiliary type	6	AMP	no
AMP-7	HNRNPA2B1::ETV1 (H9E6)	Acinar adenocarcinoma of the prostate	>11,194	AMP	
AMP-8	MTMR::MAML2 (M2M1)	Cholangiocarcinoma	5	AMP	no
AMP-9	MYB::NFIB (M11N9)	Adenoid cystic carcinoma	>100	AMP	
AMP-10	MYB::NFIB (M12N9)	Adenoid cystic carcinoma	>1800	AMP	
AMP-11	MYB::NFIB (M12N9)	Adenoid cystic carcinoma	>580	AMP	
AMP-12	MYB::NFIB (M13N9)	Adenoid cystic carcinoma	>6000	AMP	
AMP-13	MYB::NFIB (M13N9)	Adenoid cystic carcinoma	>100	AMP	
AMP-14	MYB::NFIB (M14N10)	Adenoid cystic carcinoma	>310	AMP	
AMP-15	MYB::NFIB (M14N10)	Adenoid cystic carcinoma	>5873	AMP	
AMP-16	PTPRK::RSPO3 (P1R2)	Adenocarcinoma of the hepatopancreatic ampulla, pancreato biliary type	>800	AMP	
AMP-17	SND1::BRAF (S9B9)	Pancreatic ductal adenocarcinoma	>3800	AMP	
AMP-18	TMPRSS2::ERG (T1E2)	Acinar adenocarcinoma of the prostate	>974	AMP	
AMP-19	TMPRSS2::ERG (T1E2)	Acinar adenocarcinoma of the prostate	>15,022	AMP	
AMP-20	TMPRSS2::ERG (T2E4)	Acinar adenocarcinoma of the prostate	>3432	AMP	
AMP-21	TMPRSS2::ERG (T2E4)	Acinar adenocarcinoma of the prostate	>177	AMP	
AMP-22	TMPRSS2::ERG (T1E5)	Acinar adenocarcinoma of the prostate	>11,745	AMP	
AMP-23	TMPRSS2::ERG (T2E4)	Acinar adenocarcinoma of the prostate	>2666	AMP	
AMP-24	TMPRSS2::ERG (T1E4)	Acinar adenocarcinoma of the prostate	>1207	AMP	
AMP-25	TMPRSS2::ERG (T1E4)	Adenocarcinoma of the prostate	>2000	AMP	
AMP-26	VTCN1::NRG1 (V2N2)	Cholangiocarcinoma	>12,000	AMP	
AMP-27	WHSC1L1::FGFR1 (W1F2)	Adenocarcinoma, pancreatobiliary type	>100	AMP	
OCA-1	BRD4::NUTM1 (B11N2)	CUP	>102,528	OCA	
OCA-2	EML4::ALK (E20A20)	CUP	893	OCA	
OCA-3	EML4::ALK (E6A20)	NSCLC	>6200	OCA	
OCA-4	ESR1::QK1 (E2Q5)	CUP	2857	OCA	
OCA-5	FNDC3B::PIK3CA (F3P2)	Gallbladder carcinoma	73	OCA	no
OCA-6	KIF5B::RET (K15R12)	CUP	744	OCA	no
OCA-7	RNF130::SEPT14 (R3S10)	CUP	>2153	OCA	
OCA-8	SND1::BRAF (S9B9)	Pancreatic ductal adenocarcinoma	85,887	OCA	
OCA-9	TBL1XR1::PIK3CA (T1P2)	Chordoma	531	OCA	no
OCA-10	TBL1XR1::PIK3CA (T1P2)	Colloid carcinoma of the pancreas	973	OCA	
OCA-11	TMPRSS2::ERG (T1E4)	CUP	35,966	OCA	
OCA-12	TRIM24::BRAF (T9B11)	CUP	341,016	OCA	
OCA-13	WHSC1L1::FGFR1 (W1F2)	Anaplastic carcinoma of the thyroid	>345	OCA	

* For validation primers were designed to align to either side of the specific fusion breakpoints (Appendix A). The PCR products were directly Sanger sequenced.

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
