# Peer review of "RNA-Based Detection of Gene Fusions in Formalin-Fixed and Paraffin-Embedded Solid Cancer Samples"

_cancers, 2019, doi:10.3390/cancers11091309_

Round 1
Reviewer 1 Report
All original concerns have been addressed.
There appears to be a typo (or English issue) on line 77 of the revised MS. The term 'performed analyzes' does not make sense.
Reviewer 2 Report
The authors have addressed all the points raised by reviewer 2. No further comments.
This manuscript is a resubmission of an earlier submission. The following is a list of the peer review reports and author responses from that submission.
Round 1
Reviewer 1 Report
In this manuscript, Kirchner et al., describe the results from a study in which 517 cancer patient samples were assessed by one of two different NGS-based assays designed for gene fusion detection. They detected fusions in 34 of these samples, most of which were confirmed by an orthogonal assay. They also compared turnaround times between the two assays.
Major Concerns
-The authors report sensitivity metrics; however it appears as if sensitivity assessment was not performed. It is stated that there were no false negatives, but from the description of the Results, only positive fusion cases via the NGS assays were tested by alternative assays. How then do the authors know that there were no false negatives?
-Following on from above, false positives were determined as cases for which the fusion could not be verified via RT-PCR and Sanger sequencing. Were the primers used for confirmation clinically validated? If not, it seems just as likely that Sanger results were false negative. As the authors point out, the analytic sensitivity of Sanger is generally lower than that of NGS, so for these to be truly called false positive, a confirmed negative via a validated assay would be needed.
-It needs to be made explicitly clear which fusions were deemed “false positives” (i.e. a separate table) and which methodologies, including primer sequences, failed to confirm the presence of the fusion.
-For the results deemed false positive, was there any attempt to test the sample on the other NGS assay (e.g. were the “false positive” findings via Oncomine tried on the Archer assay)?
-It is not reasonable to compare true positive rates between the assays in this study, since the assays were applied to very different patient cohorts comprised of different disease types.
Minor Concerns
-The authors refer to “fractions” of the fusions, is this meant to mean different fusion isoforms (see figure 4)?
-The use of molecular barcodes is not described accurately. Molecular barcodes are meant to bioinformatically remove PCR duplicates (i.e. identical sequencing reads), however the authors describe their use in separation of fusion transcripts. This needs to be clarified.
-Figure 2 is very difficult to interpret as some of the colors are used more than once.
-A missed opportunity in this study is comparison between the two assays. It would have been informative to test drop-outs from one assay on the other (for example, test samples with a QC score >28 via the Archer assay on the Oncomine assay).
-It would be very helpful to add more detail to Figure 1. Specifically, which steps comprise which days of the assay.
Reviewer 2 Report
- In this work, Kirchneret al. studied gene fusion detection by RNA sequencing in a cohort of 517 cases. On my opinion, the manuscript is difficult to understand. The organization is not clear and results and discussion need to be largely re-written.
- The last paragraph of the Introduction section (data) should be included in the results section.
- Why were some cases analyzed with the OCA panel and others with the AMP? What are the main differences between the NGS panels? What are the appropriate inclusion and exclusion criteria for this study (cases)? Define, described the orthogonal methods used in the study. All this information should be included in the results section.
- Please add subsections in the results section.
- In order to understand the results, it will be useful to describe the results found separately by NGS- OCA and -AMP panels.
- Table 2 should go in the supplemental information.
- Please add a table with comparison techniques: OCA vs orthogonal methods and AMP vs orthogonal methods, including specificity, sensitivity and cohen kappa values.
- Consider to add a pie with gene fusion frequency identified with each panel.